# Does the Digital Economy Promote Dietary Diversity Among Chinese Residents?

**DOI:** 10.3390/foods14223873

**Published:** 2025-11-13

**Authors:** Hao Fan, Qian Xu, Jingjing Wang, Mingming Du

**Affiliations:** 1College of Economics and Management, China Agricultural University, Beijing 100083, China; b20233110907@cau.edu.cn (H.F.); jwang010@cau.edu.cn (J.W.); 2Academy of Global Food Economics and Policy, China Agricultural University, Beijing 100083, China; 3Sydney Institute of Agricultural, School of Life and Environmental Sciences, The University of Sydney, Sydney, NSW 2015, Australia; mingming.du@sydney.edu.au

**Keywords:** dietary diversity, digital economy, food consumption, mediating effect

## Abstract

Understanding how digital transformation shapes dietary behavior is essential for evaluating nutritional transitions in developing economies. However, the mechanisms through which the digital economy impacts dietary diversity remain insufficiently explored. This study provides new empirical evidence on how digitalization influences the dietary diversity of Chinese residents. Utilizing unbalanced panel data sourced from the China Nutrition and Health Survey (CHNS), we calculate provincial digital economy indices and estimate its effects through Ordinary Least Squares (OLS) and mediating effect model. Baseline results state that the digital economy significantly improves dietary diversity. Mediating effects analyses reveal that the digital economy augments dietary diversity by boosting household income, deepening dietary awareness, and facilitating industry transformation and consumption upgrading. Moreover, heterogeneity analyses indicate that the synergistic effect between the digital economy and diet patterns varies significantly across urban–rural areas, education levels, and household living conditions. These findings offer valuable insights for other emerging economies undergoing similar digital and nutritional transitions.

## 1. Introduction

Along with improved agricultural productivity, a notable shift has occurred in the dietary patterns of Chinese residents. In the early stages of development, the major challenge was inadequate food accessibility and undernourishment, as most households primarily consumed staple grains and plant-based foods to meet basic caloric needs [1]. With rising incomes and expanding food markets, dietary preferences have gradually shifted from the goal of simply “eating enough” toward more diverse and energy-dense diets. From 1990 to 2023, a record in the China Statistical Yearbook (2024) [2], there was a marked reduction in the per capita consumption of grains (including cereals, tubers, and legumes), plummeting from 654 g/day to 330 g/day. Conversely, the appetite for meat (encompassing pork, beef, mutton, and poultry) has surged, climbing from 55 g/day to 143 g/day. In parallel, the intake of edible vegetable oil has witnessed an uptick, ascending from 16 g/day to 26 g/day [2]. These changes signify a transition from food scarcity to abundance, improving the overall nutritional status of residents. However, the rapid increase in high-fat foods has also introduced new nutritional risks, such as excessive energy intake and imbalanced dietary structures, thereby contributing to the rising prevalence of diet-related chronic diseases such as obesity, diabetes, and hypertension [3,4]. Against this background, dietary diversity has emerged as a crucial indicator of nutritional quality and health outcomes [5], reflecting not only the variety of foods consumed but also the balance between traditional staples and emerging high-energy foods. Understanding how economic, technological, and social transformations shape dietary diversity is therefore essential for evaluating the nutritional consequences of China’s ongoing dietary transition.

Food consumption patterns are shaped by diverse economic conditions across countries [6]. In China, the growth of the digital economy has become increasingly prominent since the 21st century. The China Academy of Information and Communications Technology projects that by 2023, China’s digital economy has reached 53.9 trillion yuan, constituting 42.8 percent of the GDP, a share comparable to that of the secondary industry in the overall national economy [7]. The digital economy is featured by digitization, driving industrial transformation, enhancing resource allocation, and elevating consumption upgrading [8,9]. The widespread diffusion of advanced digital technology has increased consumer choices and enriched consumption styles, particularly evident in their dietary choices. On the one hand, the surge of e-commerce and new media platforms has diversified food access channels, facilitating consumers in online shopping [10]. On the other hand, targeted promotions and precise identification on the Internet have created an isolated information environment for consumers [11], inadvertently resulting in overreliance on digital platforms for information access and potentially homogenizing individual food options. Therefore, evaluating the impact and exploring potential channels through which the digital economy influences residents’ dietary diversity is imperative.

Academic research on the dietary diversity of Chinese residents encompasses several primary domains. First, studies on the dietary quality indices have developed a series of framework to evaluate both the adequacy and diversity of food intake. Early efforts introduced the Diet Quality Index (DQI) for China [12], later expanded internationally through the DQI-I for cross-country comparisons [13]. In line with ongoing dietary transitions, subsequent tools such as the Dietary Balance Index (DBI) and the Chinese Healthy Eating Index (CHEI) were designed to align with national dietary guidelines [14,15]. These developments mark a shift from nutrient-focused evaluations to more comprehensive, culturally adapted instruments capable of capturing dietary diversity and balance in different populations. Second, a growing body of research has examined the socioeconomic and behavioral factors of dietary diversity. Urbanization, income, education, and dietary knowledge have been consistently identified as key factors shaping food consumption patterns in China [16,17,18]. More recently, digital technologies have begun to transform these dynamics. Digital nudges, such as app-based food recommendations and online platforms, are increasingly used to promote healthier and more sustainable diets, though their effects depend on user autonomy and engagement [19,20].

Despite these advances, few studies have analyzed dietary diversity through the perspective of the emerging digital economy. Existing evidence suggests that e-commerce and smartphone use can improve dietary diversity in rural China [21,22,23]. Nevertheless, most studies rely on cross-sectional data or single-channel analyses, limiting their ability to explain long-term and regional variations. Other research highlights that digitalization has accelerated industrial transformation, increased household income, and reshaped consumption behavior [24,25]. However, the mechanisms linking the digital economy to dietary diversity remain underexplored. To fill this gap, this study investigates how digital economic development influences dietary diversity among Chinese residents, integrating both regional and individual perspectives within a unified analytical framework.

This study aims to address the existing research gap through answering three key questions: How does the digital economy affect the dietary diversity of Chinese residents? What mechanisms manifest these effects? Does it facilitate a transition towards more balanced and diverse dietary patterns? Employing data from the CHNS spanning 2000 to 2011, we empirically examine the influence of digital economic development on dietary diversity. The mechanism analysis considers both regional and individual dimensions, focusing on industrial transformation and consumption upgrading at the regional level, as well as income and awareness at the individual level. We further investigate how digitalization contributes to changes in residents’ dietary patterns. The results manifest that the digital economy significantly enhances dietary diversity by increasing income and dietary awareness while promoting industrial transformation and consumption upgrading. It also drives a structural shift in the dietary patterns of Chinese residents toward more balanced and diversified consumption.

This study makes three main contributions. First, it broadens the literature by examining the impact of the digital economy on dietary diversity from a macro and long-term perspective, whereas previous studies have mainly focused on micro evidence, cross-sectional data, or specific digital tools such as smartphone use [21,22]. Second, it establishes an integrated analytical framework that simultaneously consider individual and regional mechanisms—including income growth, dietary awareness, industry transformation and consumption upgrading—to explain how digitalization reshapes food consumption. Third, it incorporates dietary pattern clustering into the empirical analysis to explore the interaction between the digital economy and dietary structures, revealing heterogeneity across urban and rural areas, education levels, and household environments. Together, these contributions provide a more comprehensive understanding of how digital transformation influences residents’ dietary diversity in China.

## 2. Hypotheses Development

### 2.1. Influence of the Digital Economy

Bennett’s Law states that with the progressive increase in individuals’ incomes and purchasing power, their food choices expand, leading to a greater diversity in food consumption [26]. Economic growth increases incomes and shifts dietary preferences from starchy staples like rice and noodles to a more diversified diets including vegetables, meat, fruits. The emergence of the digital economy as a vital paradigm influences China’s societal and economic landscape, contributing to substantial dietary diversity. The digital economy functions in three main ways. First, digital technology empowers the food manufacturing industry, thereby facilitating resources enrichment in food production [27]. By employing digital technology, production and processing enterprises can optimize processes, improve resource allocation efficiency, reduce production costs, and incentivize greater investments in research and innovation [28]. These improvements lead to lower commodity prices, thereby facilitating the accessibility of a greater range of foodstuffs for budget-constrained consumers. Second, the evolution of digital technology has spurred innovations within the sales chain, transforming the retail model from traditional offline superstores to diverse online platforms, including e-commerce, live streaming, and takeaway services [29]. By leveraging extensive internet-driven traffic, products on online platforms receive comprehensive promotion and exposure, thus significantly expanding their market reach. The increased visibility facilitates a deeper consumer understanding of commodities and aids in establishing robust product images. Consequently, these factors generate consumer interest and stimulate demand. The increased demand in turn leads to the enhancement of food options. Third, the digital economy expands consumption patterns [9], which encompass a shift from traditional offline modes of acquiring fresh food and dining at restaurants to online fresh food retail and internet-based catering services. Internet platforms offer a more diverse and expansive range of food options, regardless of temporal or geographical constraints. They also enable direct information link and delivery between producers and consumers in commercial activities [30], reducing transaction costs and increasing transaction efficiency. Considering the preceding analysis, this study posits the following hypothesis:

**Hypothesis** **1** **(H1).***The digital economy positively impacts on the dietary diversity of the population*.

### 2.2. Individual Effect of the Digital Economy

The expansion of the digital economy is a powerful engine for driving income increases. It revolutionizes how information is disseminated by means of Internet platforms. The advent of online platforms enhances job choices and aligns workers with suitable employment opportunities, thereby reducing structural unemployment [31]. Concurrently, the merging of the digital economy with productive sectors can significantly boost productivity and drive economic growth, subsequently leading to higher wages for the population [24]. From Keynes’s absolute income hypothesis, income is the principal determinant of consumption patterns, suggesting that higher income levels among residents relax their budgetary constraints, resulting in increased consumption and greater diversity, including food choices.

In the digital era, information accumulation, dissemination and coverage have significantly enhanced. Digital technologies have transformed traditional media to online platforms and burgeoning digital platforms (including online social network and live broadcast, etc.) persistently emerge. Both of them offer consumers diverse channels for acquiring information and mitigate the information asymmetry [32]. For instance, smartphone applications are creating avenues consumers to make healthy conscious dietary choices [33]. Moreover, the conceptual orientation of health diets has been extensively disseminated through digital platforms, contributing to the establishment of healthy diet cognition. Additionally, consumers are often guided by opinion leaders. Online opinion leaders who advocate healthy diets will also promote the diverse diets. Accordingly, we propose the subsequent hypothesis:

**Hypothesis** **2** **(H2).***The digital economy indirectly promotes dietary diversity through boosting individual income and enhancing dietary awareness*.

### 2.3. Regional Effect of the Digital Economy

One notable aspect of traditional industrial transformation pertains to the continuous evolution and extensive diffusion of digital technology. Specifically, the utilization of advanced software algorithms and intelligent hardware equipment lays a robust foundation for industrial transformation. The core of digitization lies in the efficient utilization and unrestricted flow of data within the industry sectors. The arrangement and employment of digital technology substantially augment the capacity to gather, integrate and analyze data within the industry [34], fostering production technology innovation and business models upgrade, thus enriching the width and depth of industry. For instance, digital technology has transformed traditional agriculture to modern digital agriculture [35]. Internal digital system effectively integrates agricultural production-related data, bolstering the efficiency of productivity and fostering the diversification of products. Consequently, consumers benefit from an expanded array of foods to diversify their diets.

Upgrading consumption signifies a transition from subsistence to development and enjoyment, which is characterized by high value, personalization and diversification of products [36]. The advent of digital era has not only reshaped consumer attitudes and patterns but also shifted the delivery of products and services. This shift has catalyzed the emergence of online consumption platforms [37], prompting consumers to adopt online consumption gradually. Online platforms offer consumer a broader array of goods, coupled with a transparent price comparison mechanism [38], enhancing convenience and choice. Concurrently, the digital economy stimulates customized consumption, leveraging big data and artificial intelligence to furnish consumers with diversified product recommendations [39], countering the homogenization and bolstering diversity. Considering the fresh produce, smartphone applications, websites and other digital platforms keep consumers free from the restriction of time and space to purchase diversified products, thereby improving their diets diversity. Consequently, we formulate the succeeding hypothesis:

**Hypothesis** **3** **(H3).***The digital economy indirectly increase dietary diversity through driving industrial transformation and promoting consumption upgrading*.

The preceding analysis of the digital economy and dietary diversity has led to the formulation of our research framework (Figure 1).

## 3. Data, Variables, and Methodology

### 3.1. Data and Variables

In our research, the dietary diversity data mainly came from CHNS, and we selected five years of data: 2000, 2004, 2006, 2009, and 2011 (The CHNS dietary data could not be matched to China Food Composition databases codes prior to 1997, and the digital economy had been gradually developing in China since the early 21st century and 2011 is the most current year to which we have access). This paper uses the nine sampled provinces or autonomous regions between 2000 and 2011: Liaoning, Heilongjiang, Shandong, Jiangsu, Henan, Hubei, Hunan, Guangxi, and Guizhou, these regions are well representative. These provinces reflect the distinctive economic development levels, industrial structures, and dietary culture of northeast, east, central and west China, which are recognized as the four typical regions in China. After data matching, cleaning, and processing, we retained the samples for the population aged over 16 years old. Finally, 13,545 unbalanced panel data were obtained. Measurements of the digital economy and other data was primarily sourced from the official website of the National Bureau of Statistics of China (NBSC), along with the China Statistical Yearbook, Statistical Yearbook of China’s provinces, China Labor Statistical Yearbook, Statistical Report on China’s Internet Development. Before conducting the empirical regressions, due to missing statistics, we utilized a Linear Interpolation Algorithm to fill in the missing values for the digital economy development of some provinces in certain years.

This study employs the Dietary Balance Index (DBI_22) as the primary measure of residents’ dietary diversity, following the established approach [14,40]. The index is designed in accordance with the Dietary Guidelines for Chinese Residents (2022) and evaluates dietary balance based on recommended intake levels across major food groups. The DBI comprises fourteen components, including cereals, vegetables, fruits, dairy products, soybeans, animal-based foods (livestock and poultry, aquatic products, and eggs), energy-providing foods (edible oils, alcoholic beverages, and sugars), condiments (converted to salt equivalents according to sodium content per 100 g), food variety, and water consumption. Each component is scored within a range of −12 to 12, with 0 indicating adherence to dietary recommendations, positive scores representing excessive intake, and negative scores denoting insufficient intake. Details on the construction of index are provided in Appendix A. As water consumption data are unavailable in the CHNS, this study constructs the DBI using the remaining thirteen components.

Additionally, explanations are warranted across certain food groups. For edible oil and salt, recorded at the household level in CHNS, individual consumption values are weighted by each person’s share of total household energy intake. The food variety (DDS) comprises twelve categories: rice products, wheat products, coarse grains and tubers, dark-colored vegetables (≥500 µg carotenoids per 100 g), light-colored vegetables (<500 µg carotenoids per 100 g), fruits, soybeans, dairy products, livestock and poultry, eggs, and aquatic products. The minimum daily intake threshold is set at 5 g for soy products and 25 g for the other food groups; a value of 0 is assigned when the threshold is met or exceeded, and −1 otherwise. Food composition data are drawn from the China Food Composition Tables (1991, 2002, and 2004 editions) compiled by the Chinese Center for Disease Control and Prevention (CDC). Observations with implausible energy intake (<1000 kcal or >3200 kcal) are excluded to ensure data consistency.

Based on the DBI, three indicators are derived: the High-Bound Score (HBS), Low-Bound Score (LBS), and Diet Quality Distance (DQD). HBS aggregates all positive scores, reflecting the extent of overconsumption. LBS sums all negative scores, indicating the degree of underconsumption. DQD, the absolute sum of all indicator scores, represents the overall imbalance of dietary intake. A DQD value of zero denotes a fully balanced diet, while higher values imply greater deviation from dietary recommendations. In this study, DQD serves as the principal measure of dietary diversity, as it captures both the magnitude and direction of imbalance within overall consumption patterns.

Provincial digital economy development level was set as the core explanatory variable in our study. This paper utilized a calculated digital economy index to measure the provincial digital economy development level and then match it with the CHNS data. To ensure data availability and accuracy, we constructed a provincial digital economy index, drawing from established indicators and methodologies in previous studies [41]. We adopted eight indicators to calculate the provincial digital economy index using a global principal component analysis (GPCA), including the business volume of post and telecommunications, year-end mobile telephone subscribers, year-end landline telephone subscribers, Internet population, Internet penetration rate, number of employees in the information-related industry, website counts, and numbers .cn domain. Through standardization and dimensional reduction, we obtained a digital economy index. Specific calculations are in the Appendix A.

This study uses two types of mediating variables, one of these is individual variables: income (household income) and dietary awareness [17]. The other is regional variables: industry transformation (one minus Industrial Structure Theil Index) and consumption upgrading (one minus Engel’s Coefficient). Industrial transformation assesses the rationality of each region’s industrial structure, specifically how resources are allocated among the three industries. Consumption upgrading denotes changes in residents’ consumption patterns, indicating improvements in both the structure and level of various expenditure types. These indicators are quantified on a scale from 0 to 1, where higher values signify a more rational industrial structure and increased consumption levels. The industry transformation variable is calculated in the Appendix A. The remaining three variables are obtained from yearbooks or CHNS.

Following previous studies on dietary behaviors [42], we introduce three types of control variables: individual, household, and provincial characteristics. The individual control variables are gender, age, marital status, highest level of education, and activity level. For household control variables, we consider household size, computer ownership, and refrigerator ownership. The provincial control variables consist of GDP per capita and total retail sales of consumer goods. For both nominal indicators, we deflate them with a base period of 2000 using the consumer price index before regressions.

### 3.2. Descriptive Statistics

Table 1 presents a comprehensive summary of the main variables used in this study. Among the dietary indicators, the mean DQD value is 39.164, with an average HBS of 15.354 and LBS of −19.010, suggesting that both excessive and insufficient food intakes coexist within the sample. The mean DDS is −4.945, indicating a moderate level of dietary diversity. The standard deviations of all dietary indicators are smaller than their respective absolute mean, suggesting less fluctuating of these data. The mean digital economy index value is 59.501, with a standard deviation of 6.963, indicating relatively stable fluctuations. The index ranges from a minimum of 48.677 to a maximum of 77.228, highlighting disparities in digital economy development across regions. We further provide regional and temporal variation for the digital economy index and the DQD in the Appendix A (see Appendix A). Additionally, Table 1 also presents descriptive statistics for other variables. Our sample is predominantly comprising middle-aged, married individuals with moderate education levels and household living conditions. These descriptive results provide a general overview of dietary diversity, digital development, and demographic characteristics, forming the basis for subsequent empirical analysis.

### 3.3. Empirical Models

This study proposes a benchmark regression model that uses OLS to investigate the impacts and underlying mechanisms of the digital economy on dietary diversity among Chinese residents. The model setting is as follows:(1)$${Diversity}_{it}=\alpha_{0}+\alpha_{1}{Dig}_{ijt}+\alpha_{2}{Control}_{ijt}+\delta_{t}+\gamma_{j}+\varepsilon_{it}$$

In the model above, *Diversity_it_* is an explained variable representing dietary diversity. Dietary diversity is primarily measured by the DQD, which accounts for both the balance and composition of food intake. This indicator incorporates the relative weights of different food groups based on individual energy consumption and reflects the extent of equilibrium across food types, thereby capturing the overall balance of dietary diversity. *Dig_it_* act as the principal explanatory variable, denoting the development degree of the sample provinces through a digital economy index. *Contorl_it_* represents the control variables at the individual, household, and provincial levels. Notations include *δ_t_* for year fixed effects, *γ_j_* for community fixed effects, and *ε_it_* for the random disturbance term, *α*_0_ is the intercept term, as well as *α*_1_ and *α*_2_ signifying the influence of the digital economy and control variables on residents’ dietary diversity.

In this study, we investigate the potential channels of the digital economy’s effect on dietary diversity through a mediating effect model. The model estimation form is shown below:(2)$${Med}_{it}=\beta_{0}+\beta_{1}{Dig}_{ijt}+\beta_{2}{Control}_{ijt}+\delta_{t}+\gamma_{j}+\varepsilon_{it}$$
where *Med_it_* represents the mediating variables including industry transformation, consumption upgrades, income, and dietary awareness. We are interested in coefficient *β*_1_ primarily, *β*_1_ denotes the effects of the digital economy on the mediating variables. The other variables are the same as those in Equation (1).

## 4. Results and Discussion

### 4.1. Benchmark Regression

The regression results derived from the benchmark model in Table 2 displays the influence exerted by the digital economy on dietary outcomes. In line with our DBI framework, we report three DBI components—DQD, HBS, and LBS—as well as the DDS, used as a complementary component within our DBI setting. Column 1 presents results for DQD, Column 2 for HBS, Column 3 for LBS, and Column 4 for DDS.

Across the columns, it is evident that the digital economy exerts beneficial effects on diet. Specifically, the coefficients on the digital economy are negative and statistically significant for DQD and HBS, indicating that digital development reduces overall dietary imbalance and excessive intake. The coefficient for LBS is negative but not statistically significant, suggesting limited effects on insufficient intake. Similarly, the estimate for DDS is positive and statistically significant at the 10% level, implying that advances in the digital economy are associated with modest improvements in dietary diversity. Taken together, these findings indicate that the diffusion of digital technology in China affects both production and consumer behavior in ways that improve dietary balance and, to a lesser extent, diversity, which is consistent with observed reality and provides preliminary validation for Hypothesis 1 of this study.

Furthermore, an examination of other control variables in the benchmark model revealed noteworthy insights. Gender shows large and significant associations (higher DQD and HBS, lower LBS), whereas its coefficient in the DDS specification is small and not significant. Age and age squared are not significant for DQD/HBS/LBS, while in the DDS regression age is positive and age squared is negative (both significant), implying an inverted-U pattern of diversity over the life cycle. The highest education level attained is associated with lower DQD and higher DDS, and it corresponds to higher LBS (i.e., less deficiency), consistent with more balanced and diverse diets among the better educated. The activity level is positively related to DQD and HBS but negatively to LBS and DDS, suggesting that individuals engaged in more strenuous physical labor tend to over-consume energy-dense items and maintain less diverse diets, which aligns with practical observations. Household computer and refrigerator ownership are strongly associated with better diets: both are linked to lower DQD (and higher LBS) and higher DDS, indicating that information access and cold storage help sustain balanced and more varied food consumption. Household size shows no significant effects. Regarding macro conditions, GDP per capita displays mixed associations across specifications and is negatively related to DDS, while total retail sales of consumer goods are not statistically significant. Overall, the implications of the control variables are consistent with realistic logic and will not be reiterated.

### 4.2. Mechanism Test

#### 4.2.1. Individual Effect: Income and Dietary Awareness

First, we assessed the impact of the digital economy on income. As shown in Table 3, Column 1, the effect of the digital economy on income is significantly positive, suggesting that its growth has contributed to an increase in total household income among Chinese residents. The expansion of the digital economy is instrumental in fostering overall economic growth through its deep integration with the tangible economy [43]. This synergy has allowed for the widespread distribution of technological benefits to the population. Moreover, digital platforms facilitate knowledge dissemination [44], thereby improving workforce quality and strengthening the alignment between the labor force and available job opportunities. This ultimately translates into increased income for residents, contributes to an overall enhancement in living standards, and augments purchasing power, as well as diversifies their dietary choices.

Moreover, we tested the mechanism of dietary awareness in individual effects. The results are presented in Column 2 of Table 3, revealing a significant positive effect, indicating that the development of the digital economy enhances the dietary awareness of the population. Accordingly, the digital economy has widened the channels through which individuals can access dietary information, broadened the reach of information dissemination, enriched residents’ knowledge of dietary matters, and contributed to the evolution of dietary awareness. Using search engines, social media, and other contemporary digital platforms [45], individuals actively seek dietary information to augment their dietary cognition. Simultaneously, while navigating the wealth of information available on the Internet, individuals passively absorb external inputs, which in turn impacts their perception of dietary consumption and improves dietary diversity. Thus, Hypothesis 2 is confirmed.

#### 4.2.2. Regional Effect: Industry Transformation and Consumption Upgrading

We proceed to examine the regional implications of industrial transformation as measured by the Industrial Structure Theil Index. The findings pertaining to the impact of the digital economy on industrial transformation are depicted in Column 3 of Table 3. The regression coefficient for the digital economy exhibits positive significance at the 1% level, indicating its capacity to drive industrial structure transformation towards rationalization. Therefore, the evolution of the digital economy alleviates resource misallocations, augments resource allocation efficiency, promotes industrial structure rationalization, and contributes to enhanced productivity in enterprises and industries. Digital technology empowers both traditional and emerging industries [46], enriching the overall industrial landscape. The enrichment improves production efficiency, diversifies product offerings, and establishes a foundation for improving residents’ dietary diversity.

Finally, we investigated the mechanism of consumption upgrading within a regional context by employing provisional Engel coefficients to represent the effect. Column 4 of Table 3 exhibits the influences of the digital economy on consumption upgrades. Our finding revealed a significant positive effect at the 5% level. Specifically, an increase in the digital economy index is associated with a decline in the Engel coefficients. This indicates that as the digital economy advances, there is a decrease in the proportion of food expenditure relative to total personal consumption expenditure, signifying a significant elevation in the population’s overall consumption level. The digital economy has substantially facilitated consumption upgrade by pathways enhancement, patterns innovation, and attitudes shaping. The digital economy has substantially enhanced the dietary structure of the population, leading to augmented diversity in dietary consumption by expanding the avenues through which the population accesses food and elevating the accessibility of a wide range of food products [47]. Moreover, the digital economy shapes dietary preferences and concepts pertaining to diverse dietary choices available to the population [48]. Consequently, the digital economy enhanced dietary diversity by facilitating consumption upgrades. The result supports Hypothesis 3.

### 4.3. Heterogeneity Analysis

#### 4.3.1. Dietary Pattern and Digital Economy

Dietary diversity serves as an essential indicator for evaluating overall dietary patterns [48]. The Chinese population is experiencing a profound transformation in food consumption, marked by increasingly diversified eating behaviors. In this section, we focus on examining how the digital economy influences residents with different dietary patterns and explore the heterogeneity of its effects under varying consumption structures. We utilized the K-means clustering algorithm to identify distinct dietary patterns within the sample population. The clustering was based on the daily average intakes of 12 food groups for each individual, as detailed in Appendix A. According to the clustering results, two major dietary patterns were identified. Overall, compared with Type II, Type I exhibits a more diversified and nutritionally balanced structure. Individuals in this group consume higher amounts of nutrient-dense foods such as vegetables, fruits, dairy products, and aquatic products, reflecting better dietary quality and greater variety in food choices.

To further examine the heterogeneity of the digital economy’s impact under different dietary patterns, we conducted separate regressions for the two clustered groups. The results are presented in Table 4. The coefficient of the digital economy is negative but statistically insignificant for Type I, indicating that digitalization exerts a limited influence on dietary quality within this group. In contrast, the coefficient is significantly negative at the 1% level for Type II, suggesting that the digital economy markedly reduces the DQD for this group. These findings imply that digital development has a stronger corrective effect on residents with relatively unbalanced diets, as the diffusion of digital technology enhances food accessibility, information acquisition, and consumption efficiency. Overall, the results confirm that the impact of digitalization on dietary improvement is heterogeneous across dietary patterns, with more pronounced benefits for individuals whose diets were previously less balanced.

#### 4.3.2. The Synergistic Effects of the Digital Economy and Dietary Patterns

Dietary behavior in China is deeply shaped by socioeconomic demographic factors such as residence, education, and household living conditions. As digital technologies increasingly penetrate daily life, their interaction with existing dietary patterns may yield differentiated outcomes across social groups. The synergistic effect between the digital economy and dietary patterns is therefore unlikely to be uniform, as disparities in digital accessibility, nutrition knowledge, and household facilities may amplify or attenuate the benefits of digitalization. In this section, we examine the heterogeneity of the synergistic effects of the digital economy and dietary patterns from three perspectives—urban versus rural residence, education level, and refrigerator ownership—to uncover how contextual and household differences shape the dietary gains derived from digital transformation.

We first analyzed the heterogeneity of the synergistic effects of the digital economy and dietary patterns across urban and rural areas. As reported in Panel A of Table 5, the coefficients of the digital economy are significantly negative in both subsamples, indicating that digital development effectively reduces dietary imbalance among both urban and rural residents. The interaction term between the digital economy and Type I is significantly positive in urban areas but insignificant in rural areas. This suggests that within cities, the digital economy exerts a stronger corrective effect on residents with less balanced dietary patterns, helping those with initially poorer diets to achieve better nutritional outcomes. In contrast, the absence of a significant interaction in rural areas implies that the digital economy exerts a relatively uniform influence across different dietary groups, possibly because digital access and consumption channels are still developing and have not yet led to differentiated behavioral responses among rural residents [49].

We further investigated the heterogeneity across groups with different education levels. The results are presented in Panel B of Table 5. The sample was divided according to the mean years of schooling, with individuals above the mean classified as the high-education group and those below the mean as the low-education group. The coefficients of the digital economy are significantly negative in both groups, indicating that digital development contributes to reducing dietary imbalance regardless of educational attainment.

However, the interaction term between the digital economy and Type I is positive and statistically significant in the low-education group, while it is smaller and insignificant in the high-education group. This finding suggests that for residents with lower education levels, the digital economy plays a stronger role in improving dietary balance among those with less healthy dietary structures. One possible explanation is that individuals with lower educational attainment tend to have more limited access to nutrition information and fewer opportunities for health education, making them more responsive to the information and convenience effects brought by digital technologies [25]. In contrast, among highly educated residents, the marginal effects of digitalization are weaker, as these individuals already possess greater nutrition awareness and more diversified consumption choices.

Finally, we examined the heterogeneity across households with and without refrigerators. The results are presented in Panel C of Table 5. The coefficients of the digital economy are significantly negative in both groups, suggesting that digital development helps reduce dietary imbalance regardless of household storage conditions. The interaction term between the digital economy and Type I is positive and statistically significant for households owning a refrigerator, whereas it is smaller and insignificant for those without one. This indicates that among households with refrigeration facilities, the digital economy exerts a stronger corrective effect on individuals with less balanced diets. A plausible explanation is that access to refrigeration extends the freshness and variety of perishable foods, enabling households to better utilize the convenience and diversity offered by digital food purchasing and distribution channels. In contrast, for households without refrigerators, limited storage capacity constrains the potential benefits of digitalization in improving dietary balance, leading to weaker interactive effects [50,51].

## 5. Robustness Test

### 5.1. Replacement of Explained Variable

To minimize potential bias arising from the selection of the dependent variable, and in line with the existing methodology [52], we substituted the dependent variable with the entropy (E) and Simpson index (SI) of dietary diversity. These two indicators were calculated by assessing the distribution of food consumption items, incorporating category as shares. The calculation methods are provided in the Appendix A. The regression results with E and SI as explained variable are presented in Columns 1 and 2 of Table 6. The estimation result reveals a statistically significant positive impact of the digital economy index on E and SI at a significance level of 5%. Its coefficient is relatively small due to the magnitude. This finding aligns with the findings of the benchmark regression, thereby corroborating our results robustness. Specifically, the finding underscore that the digital economy contributes positively to dietary diversity.

### 5.2. Remeasurement of Core Explanatory Variable

Additionally, considering the calculation method of core explanatory variable can also influence the model results, we have employed the entropy method to re-measure the degree of digital economy development across all selected provinces. This approach aims to enhance the robustness of our research. The results from the regression analyses using the novel digital economy index as the core explanatory variable are displayed in Column 3 of Table 6. The digital economy index still significantly positively influences DQD, indicating that the digital economy enhanced residents’ dietary diversity. Though their coefficients are smaller than previous results, the consistent positive influence substantiates the robustness of our study’s findings.

### 5.3. Quantile Model Regressions

The study employed a quantile regression using the method of moments to investigate the influence of digital economy development on dietary diversity across different quantiles [53]. The quantile regression results presented in Table 7 report both the scale equation and the conditional quantiles (*θ* = 0.25, 0.5, and 0.75) for the DQD, with all controls, year, and community fixed effects included. In the scale equation, the coefficient of the digital economy is significantly negative, indicating that digital development reduces the dispersion of dietary imbalance across individuals, thereby narrowing the dietary gap. At the conditional quantile level, the digital economy shows a negative but statistically insignificant coefficient at 0.25, while the effects are significantly negative at 0.5 and 0.75. The estimated coefficients become more negative as the quantile increases, implying that the impact of the digital economy is stronger among residents with poorer dietary balance and diversity (higher DQD values). These findings are consistent with the prior heterogeneity analysis and further confirm the robustness of our results.

### 5.4. Endogeneity Test

This study addresses two primary endogeneity issues. First, sample selection bias: despite utilizing a large dataset from a longitudinal study, the research faces the challenge of self-selection, where households with greater dietary diversity, often associated with stronger economic capabilities, may relocate to areas with a more advanced digital economy. Second, the omitted variable problem: although individual, household, and provincial variables are controlled for, and year and community fixed effects are incorporated into the regressions, dietary diversity is influenced by various factors including personal food preferences, household agricultural practices, the food consumption environment, and regional food culture. Consequently, potential omitted variable bias remains a concern.

To mitigate these endogeneity concerns, we primarily employ the Heckman two-stage model and instrumental variable (IV) regression. In the Heckman two-stage model, we determine the mean DQD for each community based on a single sample’s location. We then compare each sample’s DDS to this community mean, assigning a value of 0 for scores above the mean and 1 for those below, thereby creating dummy variables (DQD dummy) for the first stage of the Heckman model. Additionally, we calculate the mean DQD values (DQD mean) for all communities, excluding respondent self, and incorporate these into the first stage of the Probit regression. This allows us to compute the Inverse Mills Ratio (IMR) to be included in the second stage regression. The results are presented in Columns 1 and 2 of Table 8. The significance of the IMR suggests that there is a sample selection issue within the study sample. However, the coefficient of the digital economy remains significantly positive, indicating that the findings are robust even when considering sample selection bias.

Additionally, we proposed that the number of letters in sample provinces in 1990 can be used as an appropriate IV for the digital economy index. For an instrumental variable to be effective, it must correlate with the primary explanatory variable and possess exogeneity. In this case, the number of letters indicates the development of the traditional local postal and telecommunications infrastructure, which serves as foundational support for the evolution of the digital economy. The rapid progression of China’s digital economy can be attributed to its preeminent position in the establishment of information and communication networks. This achievement is intricately linked to China’s pioneering and sophisticated postal and telecommunications infrastructure. Notably, the deployment of fiber-optic cables within the postal and telecommunications sector has served as the foundational hardware platform underpinning the advancement of digital technologies. Thus, it maintains a close association with the digital economy and serves as a viable IV. Nonetheless, the data structure of this study comprised panel data, and the number of letters in 1990 was cross-sectional and lacked a temporal dimension. To address the temporal limitation, we drew inspiration from the established approach [54]. We presented a time-variant variable to formulate an interaction term that serves as an IV. The interaction term combines the number of internet users in the country from the previous year with the 1990 letter count for each sampled province. These newly constructed interaction terms were applied as instrumental variables to evaluate the provincial digital economy index for each year.

In Table 8, we employed two-stage least squares (2SLS) method for IV regression. Our findings demonstrate that the instrumental variable exhibits a significantly positive relationship with the digital economy at a 1% level in the first regression step. Proceeding to the second regression step, the digital economy index maintained a negative relationship with the DQD achieving a 5% significance level. The under-identification test yielded a *p*-value of 0.000 for the Kleibergen-Paap rk LM statistics, indicating no issue with instrumental variable under-identification. In the weak identification test, the Cragg-Donald Wald F statistic, which measures the robustness of the instrumental variable, stands at 20,830.38, surpassing the critical value 16.38 set at the 10% significance level by the Stock-Yogo weak identification test. This result indicates the absence of weak IV identification for the instrumental variable. The findings of the benchmark regression remain robust after employing the IV method to mitigate endogeneity concerns.

## 6. Conclusions

### 6.1. Research Contributions

Our research examines how the macro-level digital economy influences residents’ micro-level dietary diversity utilizing two complementary datasets. The study establishes a comprehensive framework to evaluate the effects of digitalization from an innovative perspective. We also analyze the disparities in the digital economy’s impact across the urban–rural divide, education levels, and household living conditions, and identify the mechanisms through which it shapes dietary diversity. In addition, we explore how digitalization contributes to changes in residents’ dietary patterns.

The main findings are as follows. First, the benchmark regression results state that the digital economy significantly improves dietary diversity among Chinese residents. This relationship remains robust after applying instrumental variable estimation, replacing dependent and independent variables, and performing quantile regression. This result aligns with recent studies showing that digitalization and online food platforms are reshaping consumer dietary practices worldwide [55]. However, other research has found that digital food marketing through delivery apps often promotes unhealthy snacks and processed foods, which may counteract efforts to encourage balanced diets and contribute to public health challenges [56]. Second, mechanism analysis suggests that the digital economy enhances income, increases dietary awareness, refines industrial structure, and promotes consumption upgrading, thereby improving dietary diversity. Third, extended tests demonstrate that digitalization promotes a transition in Chinese residents’ diets. Heterogeneity analysis further shows that the interactive effects of the digital economy and dietary patterns vary significantly across urban and rural areas, education levels, and household living conditions. The positive effect of the digital economy is particularly evident among individuals with less balanced diets across the urban, low-education level and households with refrigerators categories. This highlights the critical role of nutrition education and food accessibility. These findings are consistent with international evidence emphasizing the influence of food environments and accessibility [50,52].

Based on the results, several policy implications can be derived. First, it is advisable for China’s agricultural department to prioritize the integration of digital technologies into agri-food systems to enhance food supply diversity and quality. This includes expanding rural e-commerce platforms and digital logistics networks that facilitate the circulation of fresh and nutritious food between urban and rural areas, thereby reducing regional disparities in dietary access. Second, the government could encourage digital transformation within food production and distribution sectors. Supporting digital agriculture, smart supply chain management, and data-driven market matching can promote the transition from traditional to modern production models while improving efficiency and sustainability. Third, public policies could focus on digital nutrition education. Developing mobile applications, online platforms, and AI-based personalized diet tools can improve residents’ nutritional awareness and guide healthier food choices, particularly among rural and low-education groups. Finally, it would be prudent for policymakers to strengthen data governance and equitable access to digital resources, ensuring that digital transformation contributes not only to food innovation but also to inclusive food security and public health improvement.

### 6.2. Limitations and Future Research

Our research is subject to several limitations. First, the dietary diversity data are drawn from the CHNS covering 2000 to 2011, which might not fully reflect recent developments in dietary behavior. We acknowledge this limitation due to the unavailability of updated, nationally representative micro-level data. Nevertheless, this period coincides with two critical transitions in China: the rapid expansion of the digital economy and the structural transformation of residents’ diets from staple-based consumption to more diverse and nutrient-rich patterns. Examining this formative phase provides valuable insights into how digitalization initially reshaped food consumption and laid the groundwork for subsequent transformations. Moreover, the analytical framework remains applicable to other developing countries currently experiencing similar digital and nutritional transitions. Second, the construction of the digital economy index is constrained by the availability of provincial indicators. Subsequent research can enrich the index with additional, fine-grained measures (e.g., platform penetration, digital logistics, mobile payment intensity) to improve coverage and comparability. Third, while our mechanism analysis considers income, dietary awareness, industrial transformation, and consumption upgrading, further research could examine evolving channels associated with post-2010 technologies—such as cold-chain expansion, on-demand delivery, and new media exposure—to deepen understanding of how digitalization shapes dietary diversity.

## Figures and Tables

**Figure 1 foods-14-03873-f001:**
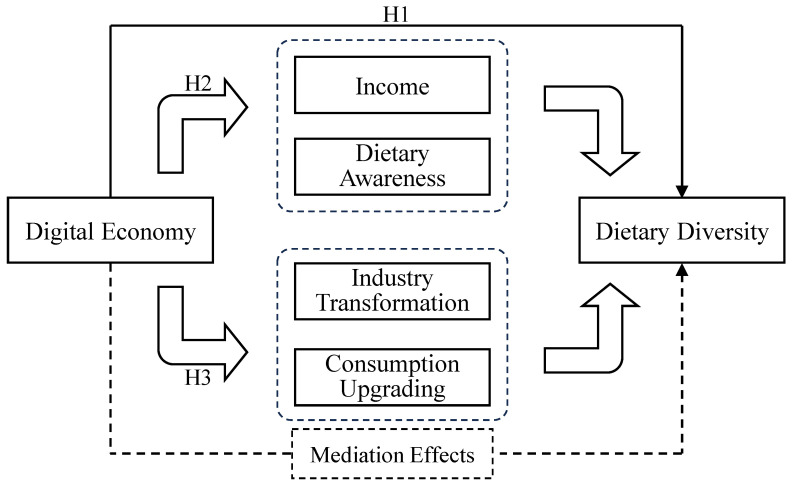
The research framework of the digital economy on dietary diversity.

**Table 1 foods-14-03873-t001:** Summary statistics.

Variable	Mean	SD	Min	Max
Dietary diversity				
Dietary quality distance (DQD)	39.164	8.674	9	71
High-bound score (HBS)	15.354	5.100	0	34
Low-bound score (LBS)	−19.010	9.471	−50	0
Food variety (DDS)	−4.945	1.726	−12	0
Digital economy index	59.501	6.963	48.677	77.228
Individual characteristics				
Gender (male = 1, female = 0)	0.554	0.498	0	1
Age	45.835	14.146	16	94
Marital status	2.003	0.500	1	5
Highest education level attained	2.703	1.272	1	6
Activity level	2.228	1.132	1	5
Household characteristics				
Computer owned (dummy)	0.311	0.463	0	1
Refrigerator owned (dummy)	0.772	0.420	0	1
Household size	3.082	1.266	1	10
Provincial characteristics				
GDP per capita (logarithm)	9.624	0.559	7.923	10.760
Total retail sales of consumer goods (logarithm)	8.100	0.762	5.840	9.494
Mediating variables				
Household income (1000 CNY)	29.104	29.399	0.1	396.4
Dietary awareness (dummy)	0.222	0.415	0	1
Industry transformation	0.729	0.149	0.167	0.906
Consumption upgrading	0.598	0.056	0.373	0.686

Notes: 1. Marital status: The question in CHNS is “What is your marital status?”, the answers are: 1–5 are never married, married, divorced, widowed, separated, respectively. 2. Highest education level attained: The question in CHNS is “What is the highest level of education you have attained?”, the answers are: 1–6 are graduated from primary school, lower middle school degree, upper middle school degree, technical or vocational degree, university or college degree, master’s degree or higher, respectively. 3. Activity level: The answers in CHNS are: 1–6 are very light physical activity, light physical activity, moderate physical activity, heavy physical activity, very heavy physical activity, no working ability, respectively. 4. Provincial data are sourced from the National Bureau of Statistics of China. 5. Dietary awareness: The question in CHNS is “Do you know about the Chinese Pagoda or the Dietary Guidelines for Chinese residents?”, the answer is no = 0, yes = 1.

**Table 2 foods-14-03873-t002:** Dietary diversity and digital economy.

Variables	(1)	(2)	(3)	(4)
	DQD	HBS	LBS	DDS
Digital economy	−0.125 **	−0.114 ***	−0.019	0.019 *
	(0.053)	(0.037)	(0.064)	(0.011)
Gender	1.105 ***	0.226 ***	−0.946 ***	−0.007
	(0.102)	(0.070)	(0.125)	(0.020)
Age	−0.037	−0.029	0.025	0.015 ***
	(0.028)	(0.019)	(0.031)	(0.006)
Age squared	0.044	0.043 **	−0.027	−0.016 ***
	(0.030)	(0.020)	(0.033)	(0.006)
Marital status	0.034	−0.124	−0.297 *	−0.061 **
	(0.148)	(0.095)	(0.173)	(0.029)
Highest education level attained	−0.457 ***	−0.064	0.383 ***	0.111 ***
	(0.067)	(0.044)	(0.080)	(0.013)
Activity level	0.542 ***	0.132 **	−0.566 ***	−0.110 ***
	(0.087)	(0.057)	(0.101)	(0.017)
Computer owned	−0.968 ***	−0.097	0.926 ***	0.236 ***
	(0.207)	(0.137)	(0.243)	(0.041)
Refrigerator owned	−1.261 ***	−0.197	1.136 ***	0.317 ***
	(0.222)	(0.146)	(0.250)	(0.045)
Household size	−0.029	−0.001	−0.095	0.004
	(0.080)	(0.055)	(0.092)	(0.015)
GDP per capita	2.482 **	−2.703 ***	−5.990 ***	−1.069 ***
	(1.263)	(0.860)	(1.504)	(0.261)
Total retail sales of consumer goods	2.859	1.605	−1.167	−0.070
	(1.943)	(1.186)	(2.283)	(0.385)
Constant	0.971	35.677 ***	49.141 ***	4.195
	(13.861)	(9.750)	(16.990)	(2.845)
Year	Yes	Yes	Yes	Yes
Community	Yes	Yes	Yes	Yes
N	13,545	13,545	13,545	13,545
R^2^	0.345	0.153	0.235	0.343

Notes: Robust standard errors in parentheses, clustered by household. * *p* < 0.1, ** *p* < 0.05, *** *p* < 0.01.

**Table 3 foods-14-03873-t003:** Mechanism test.

Variables	(1)	(2)	(3)	(4)
	Household Income	Dietary Awareness	Industry Transformation	Consumption Upgrading
Digital economy	0.680 ***	0.007 *	0.005 ***	0.001 **
	(0.219)	(0.003)	(0.001)	(0.000)
Controls	Yes	Yes	Yes	Yes
Year	Yes	Yes	Yes	Yes
Community	Yes	Yes	Yes	Yes
N	13,545	11,778	13,545	13,545
R^2^	0.354	0.218	0.922	0.861

Notes: * *p* < 0.1, ** *p* < 0.05, *** *p* < 0.01.

**Table 4 foods-14-03873-t004:** Two clustering dietary pattern and digital economy.

Variables	(1)	(2)
	DQD	DQD
	Type I	Type II
Digital economy	−0.063	−0.223 ***
	(0.120)	(0.058)
Controls	Yes	Yes
Year	Yes	Yes
Community	Yes	Yes
N	3908	9637
R^2^	0.299	0.285

Notes: *** *p* < 0.01.

**Table 5 foods-14-03873-t005:** Dietary pattern and digital economy: heterogeneity analysis.

Panel A. Grouped by Urban/Rural	DQD	DQD
Variables	Urban	Rural
Digital economy	−0.182 **	−0.224 ***
	(0.079)	(0.074)
Type I (dummy)	−9.240 ***	−5.480 ***
	(1.959)	(1.971)
Dig * Type I	0.063 *	0.023
	(0.033)	(0.032)
Controls	Yes	Yes
Year	Yes	Yes
Community	Yes	Yes
N	6329	7216
R^2^	0.352	0.367
**Panel B. Grouped by Education Level**	**DQD**	**DQD**
**Variables**	**Low**	**High**
Digital economy	−0.221 ***	−0.174 **
	(0.066)	(0.078)
Type I (dummy)	−8.820 ***	−7.506 ***
	(1.802)	(1.934)
Dig * Type I	0.065 **	0.041
	(0.029)	(0.032)
Controls	Yes	Yes
Year	Yes	Yes
Community	Yes	Yes
N	7205	6340
R^2^	0.393	0.377
**Panel C. Grouped by Refrigerator Owned**	**DQD**	**DQD**
**Variables**	**With Refrigerator**	**Without Refrigerator**
Digital economy	−0.167 **	−0.277 ***
	(0.065)	(0.106)
Type I (dummy)	−9.148 ***	−7.386 *
	(1.513)	(4.274)
Dig * Type I	0.066 ***	0.062
	(0.025)	(0.074)
Controls	Yes	Yes
Year	Yes	Yes
Community	Yes	Yes
N	10,454	3091
R^2^	0.376	0.383

Notes: * *p* < 0.1, ** *p* < 0.05, *** *p* < 0.01.

**Table 6 foods-14-03873-t006:** Remeasurement of explained and core explanatory variables.

Variables	(1)	(2)	(3)
	E	SI	DQD
Digital economy	0.003 **	0.001 **	
	(0.002)	(0.000)	
Digital economy (entropy)			−0.059 **
			(0.027)
Controls	Yes	Yes	Yes
Year	Yes	Yes	Yes
Community	Yes	Yes	Yes
N	13,545	13,545	13,545
R^2^	0.325	0.283	0.345

Notes: ** *p* < 0.05.

**Table 7 foods-14-03873-t007:** Quantile regressions for dietary diversity and digital economy.

Variables	(1)	(2)	(3)	(4)
	Scale	θ=0.25	θ=0.5	θ=0.75
Digital economy	−0.099 ***	−0.041	−0.126 ***	−0.211 ***
	(0.025)	(0.048)	(0.042)	(0.047)
Controls	Yes	Yes	Yes	Yes
Year	Yes	Yes	Yes	Yes
Community	Yes	Yes	Yes	Yes
N	13,545	13,545	13,545	13,545

Notes: *** *p* < 0.01.

**Table 8 foods-14-03873-t008:** Endogeneity tests results.

Variables	(1)	(2)	(3)	(4)
	Heckman Two-Stage Model		IV_2SLS	
	DQD Dummy	DQD	Digital Economy	DQD
Digital economy	−0.004	−0.124 ***		−0.166 **
	(0.010)	(0.032)		(0.070)
DQD mean	0.056 ***			
	(0.005)			
IMR		−6.782 ***		
		(0.059)		
IV			0.010 ***	
			(0.000)	
Controls	Yes	Yes	Yes	Yes
Year	Yes	Yes	Yes	Yes
Community	Yes	Yes	Yes	Yes
F-statistics			7457.52	
N	13,500	13,500	13,545	13,545
R^2^/Pseudo R^2^	0.030	0.723		0.028
Likelihood ratio	−9072.56			
Under identification				820.68
Weak identification				20,830.38

Under identification reports Kleibergen-Paap rk LM statistic. Weak identification reports Cragg-Donald Wald F statistic. The value of the stock-Yogo weak ID test critical value at 10% maximal IV size is 16.38. Notes: ** *p* < 0.05, *** *p* < 0.01.

## Data Availability

The data supporting this study are from the CHNS database (https://chns.cpc.unc.edu/, accessed on 18 October 2023), which can be downloaded via official application on the website, and the authors have no right to distribute the data independently.

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
