# Peer review of "Does the Digital Economy Promote Dietary Diversity Among Chinese Residents?"

_foods, 2025, doi:10.3390/foods14223873_

Round 1

Reviewer 1 Report

Comments and Suggestions for Authors

Dear Editors,

Dear Authors,

The overall idea of the paper is very interesting, examining the link between digital economy and dietary diversity. However, there are some methodological decisions that I find concerning and Authors could detail/reconsider:

Increased intake of certain food groups is hardly an "enrichment" of dietary diversity, especially if there is reduction in consumption in others. Undernourishment replaced by overeating is hardly a positive development, regardless of the subjective perception for well-being. Hence, previous poor food accessibility should be described on its own and it would be valuable Authors to delve more into this part of the question.

In the methodology, as it is now, all groups have same weight for calculating the diet diversity and no actual quantities are taken into account, so the DDS will not account a overconsumption of fast food, cakes (that are not sweets?), and sweetened beverages (placed in others). Same goes for red meats and poultry, placed in a single group. Lumping together processed foods that contain large amounts of carbs, fats and proteins will only cloud the  real diet diversity. If those could be separated in more specific groups or steps to measure the weight of the groups is added, I think it could work. I would recommend adding the dietary pattern clustering into the methodology and working with it from the start and not in the further analyses as it is highly likely Type 1 to prevail in rural sample where affect of digital economy would be lower. On the other hand, as Chinese traditional cuisine prioritize fresh produce, it is important to examine the education, refrigerator ownership and other control variables in regard to this clustering (or cite other authors, if it is already done). Taking into account all these, I think, Authors would produce more comprehensive papers with more valuable contribution to diet recommendations of Chinese people.

In technical terms: most of the tables lack captions explaining the error bars, asterisks, etc.

See those and other remarks in the file.

Reviewer 2 Report

Comments and Suggestions for Authors

Dear all, congratulations on the manuscript. It presents:

A current and relevant topic—the interface between the digital economy and food diversity is innovative and underexplored;
A solid empirical basis—uses the CHNS panel and applies robust econometric models (OLS, mediation, IV, quantile);
Consistent results—supported by robustness and heterogeneity tests;
Clear theoretical contributions—identifies mechanisms (income, food awareness, industrial transformation, consumption upgrading).

The following are suggestions for improvement:
1. The justification for the study could be improved in the introduction; even though the contributions are highlighted, they seem broad and overly general.
2. The text is understandable, but there are long and repetitive passages; it could be revised for conciseness and fluidity (especially in the introduction and discussion).
3. Data update: the data go back to 2011; the article itself acknowledges this limitation. It is suggested to better justify why the period is still relevant or include more recent data (e.g., 2015–2020, if available).
4. Originality and international comparison. The discussion could compare the results with other emerging countries, reinforcing the global scope.
5. Improve visualization, include graphs or maps that illustrate regional differences in the digital economy index and dietary diversity.
6. Practical implications: provide more detail on how public policies could integrate digital transformation and food security (e.g., rural e-commerce, nutrition education apps).
7. Literature review, update references on the digital economy and food post-2020 (particularly in Food Policy, Appetite, Technological Forecasting & Social Change).

Round 2

Reviewer 1 Report

Comments and Suggestions for Authors

I commend the Authors' efforts to improve the paper and almost redone the study. I really hope their findings will be taken into consideration in further policy making.